

# Biochemical and biophysical characterisation of immunoglobulin free light chains derived from an initially unbiased population of patients with light chain disease

Rebecca Sternke-Hoffmann[1], Amelie Boquoi[2], David Lopez Y. Niedenhoff[2], Florian Platten[3], Roland Fenk[2], Rainer Haas[2] and Alexander K. Buell[1,4]

[1] Institute of Physical Biology, Heinrich-Heine Universität Düsseldorf, Düsseldorf, Germany
[2] Department of Hematology, Oncology and Clinical Oncology, Heinrich-Heine Universität Düsseldorf, Düsseldorf, Germany
[3] Condensed Matter Physics Laboratory, Heinrich-Heine Universität Düsseldorf, Düsseldorf, Germany
[4] Department of Biotechnology and Biomedicine, Technical University of Denmark, Lyngby, Denmark

Corresponding authors
Rainer Haas,
haas@med.uni-duesseldorf.de
Alexander K. Buell, alebu@dtu.dk

## ABSTRACT

In light chain (LC) diseases, monoclonal immunoglobulin LCs are abundantly produced with the consequence in some cases to form deposits of a fibrillar or amorphous nature affecting various organs, such as heart and kidney. The factors that determine the solubility of any given LC in vivo are still not well understood. We hypothesize that some of the biochemical properties of the LCs that have been shown to correlate with amyloid fibril formation in patients also can be used as predictors for the degree of kidney damage in a patient group that is only biased by protein availability. We performed detailed biochemical and biophysical investigations of light chains extracted and purified from the urine of a group of 20 patients with light chain disease. For all samples that contained a sufficiently high concentration of LC, we quantified the unfolding temperature of the LCs, the monomer-dimer distribution, the digestibility by trypsin and the formation of amyloid fibrils under various conditions of pH and reducing agent. We correlated the results of our biophysical and biochemical experiments with the degree of kidney damage in the patient group and found that most of these parameters do not correlate with kidney damage as defined by clinical parameters. However, the patients with the greatest impairment of kidney function have light chains which display very poor digestibility by trypsin. Most of the LC properties reported before to be predictors of amyloid formation cannot be used to assess the degree of kidney damage. Our finding that poor trypsin digestibility correlates with kidney damage warrants further investigation in order to probe a putative mechanistic link between these factors.

## INTRODUCTION

The formation of insoluble aggregates by proteins can be associated with a broad range of human disorders, many of which are neurodegenerative in nature (*Dobson, 2003*; *Westermark et al., 2005*; *Knowles, Vendruscolo & Dobson, 2014*). Protein deposition in other organs than the central nervous system can also lead to a wide variety of diseases. In the case of the deposition of ordered fibrillar aggregates (amyloid fibrils) affecting various organs the related disorders are known as systemic amyloidoses (*Wechalekar, Gillmore & Hawkins, 2016*).

A particular class of such protein deposition disorders is represented by the light chain diseases, which are characterized by the occurrence of monoclonal free light chains in blood and urine (*Edelman & Gally, 1962*). In general, the monoclonal free light chains of either kappa or lambda isotype are the secreted product of monoclonal plasma cells residing in the bone marrow. Dependent on the type of the underlying B-cell disorder, they may reflect a monoclonal gammopathy of uncertain significance (MGUS), a smoldering or full blown multiple myeloma (MM) with exclusive (Bence Jones MM) or substantial light chain secretion in addition to the complete immunoglobulin. Each light chain protein has a unique amino acid sequence which is determined by somatic recombination and various mutations (*Sakano et al., 1979*). The amino acid sequence, together with potential post-translational modifications and in interplay with the local conditions in the organism, such as local pH or presence of proteases, determines the in vivo behavior of the light chain. The sequence diversity translates into a diverse clinical picture, as far as organ involvement and severity of organ damage are concerned (*Enquist et al., 2007*). Aggregation of light chains and fragments of low solubility can lead to different diseases, such as light chain deposition disease (LCDD), where the LC forms amorphous aggregates (*Sanders & Booker, 1992*; *Buxbaum & Gallo, 1999*), and the AL-amyloidosis, where the LC forms amyloid fibrils (*Glenner, Ein & Terry, 1972*).

At present, it is not possible to decide upon first diagnosis of a patient with light chain disease, whether the particular monoclonal light chain found in their blood at increased concentration is prone to form amyloid fibrils or amorphous deposits, or remain soluble and get excreted quantitatively through the urine. Even if the sequence of the particular light chain is known, its solubility inside the organism cannot currently be predicted. Therefore, there is a need both for increased mechanistic understanding of LC deposition and for easy, rapid and reliable diagnostic procedures that are able to assess the potential of a given light chain to cause damage through deposition.

Much work has been carried out in recent years on the study of the biophysical properties of LCs and to correlate the results with, in particular, their amyloid fibril formation in vivo (*Arosio et al., 2012*; *Blancas-Mejía et al., 2015*; *Andrich et al., 2017*; *Brumshtein et al., 2018*; *Weber et al., 2018*). Most of these studies have relied on the *a priori* knowledge that a given light chain forms, or not, amyloid fibrils in vivo. In this study, we set out to test the hypothesis that (some of) the biochemical and biophysical properties of light chains previously reported specifically in the context of amyloidosis also correlate with the severity of general light chain disease symptoms, in particular impairment of kidney function. In

our study, we examined 20 patients presenting with light chains in their urine. Most of them had multiple myeloma with a large variety of symptoms. For our in vitro assessment, we isolated protein from the urine of these patients, where the proteins appear in high concentration, when the production is significantly increased and the ability to reabsorb the filtered proteins is exceeded, and characterized the samples in detail with biochemical and biophysical methods. We did not apply any particular selection criteria to include patients into the study, but excluded those samples that did not contain a sufficient concentration of light chains.

## MATERIALS & METHODS

### Patients

The study described in this manuscript has been reviewed and approved by the ethics committee of the university hospital Düsseldorf and all patients of whom samples were used in the study have signed an informed consent (study number 5926R and registration ID 20170664320). The in vitro studies were performed using protein isolated from urine samples of 20 patients (6 females, 14 males, median age 61.5 years with a range between 45 and 76 years) with multiple myeloma of various subtypes and one patient with amyloidosis as detailed in Table 1 (patient characteristics). For the majority of patients –P008 and 009 had a cast nephropathy—a histopathological examination of the kidney was not available since the corresponding invasive diagnostic procedure was not necessary for the therapy decision-making process as they were diagnosed according to IMWG criteria. Thus, there was no initial bias in the selection of patients, as the common denominator for inclusion into the study was solely the presence of a monoclonal light chain in the peripheral blood and in the urine, while the type of ligt chain disease (MM vs. Amyloidosis) did not play a role.

As far as the type of light chain is concerned, 14 patients with Kappa and 6 patients with Lambda type chain were part of the study. There was a large variation with regard to the light chain concentration in the serum, with concentrations between 2.2 mg/l and 11,000 mg/l. Two of our patients presented with renal insufficiency and required dialysis, while four of them had no signs of functional renal impairment. We divided the patients into three different groups (I, II, III) according to their chronic kidney disease (CKD)-stage at the time of diagnosis. Groups I corresponds to stages 1 and 2, group II to stage 3 and group III to stages 4 and 5. At the time when the urine samples were collected, 17 patients were diagnosed *de novo*, whereas three patients with multiple myeloma had received induction therapy with bortezomib, cyclophosphamide and dexamethasone. The duration of their disease at the time of the examination was therefore relatively short with a median of 6 weeks varying from 4 to 12 weeks.

### Experimental methods

Detailed descriptions of the methods are provided in the supplemental materials.

#### *Sample preparation*

The protein content of a 24 h urine collection was precipitated by ammonium sulfate (70% saturation) and the light chains were purified by size-exclusion chromatography on

Table 1 **Patient Characteristics at the time of examination.** The patient groups are given according to CKD: Group I good Stage 1+2 $n = 9$ (P004, 007, 010, 011, 012, 014, 017, 018, 020), Group II intermediate Stage 3 $n = 6$ (P001, 005, 013, 015, 016, 019) and Group III bad Stage 4+5 $n = 5$ (P002, 003, 006, 008, 009).

| | Age | Gender | Disease type | Subtype | Subtype | Time of sample | FLC Serum (mg/l) | FLC Urine (mg/l) | TPU (g/24 h) | Kidney function creatinin (mg/dl) | GFR-CKD-EPI (ml/min) | CKD Stadium | Kidney histology | Kidney function recovered? (Improvement = 30%; yes/no) | Kidney function recovered? Creatinine (mg/dl) | chronic/acute kidney injury |
|---|---|---|---|---|---|---|---|---|---|---|---|---|---|---|---|---|
| P001 | 47 | male | MM | IgG | Lambda | at diagnosis | 3,750 | 1,060 | 13 | 1,3 | 50 | 3 | Unknown | Yes | 0,9 | c |
| P002 | 76 | male | MM | – | Lambda | after 1 course of therapy | 4,250 | unknown | 0,3 | Dialysis dependent | <10 | 5 | Unknown | No | Dialysis dependent | a |
| P003 | 52 | male | MM | IgG | Lambda | after 1 course of therapy | 2,420 | unknown | 0,3 | Dialysis dependent | 15 | 4 | Unknown | No | Dialysis dependent | a |
| P004 | 72 | male | MM | IgG | Kappa | at diagnosis | 5,280 | 7,650 | 4 | 1,1 | 90 | 1 | Unknown | Yes | 0,9 | c |
| P005 | 65 | female | MM | IgG | Kappa | at diagnosis | 1,250 | 6,140 | 3 | 1,2 | 48 | 3 | Unknown | NA | acute kidney injury | c |
| P006 | 66 | female | MM | IgG | Kappa | at diagnosis | 2,460 | 6,880 | 3 | 1,9 | 27 | 4 | Unknown | Yes | 1,3 | c |
| P007 | 54 | male | MM | IgG | Kappa | at diagnosis | 898 | unknown | 1 | 1 | 83 | 2 | Unknown | No | 1,3 | a |
| P008 | 47 | female | MM | – | Kappa | at diagnosis | 1,370 | 3,990 | 1,5 | 2,1 | 28 | 4 | cast nephropathy | Yes | 1,5 | c |
| P009 | 48 | male | MM | – | Lambda | at diagnosis | 2,590 | unknown | 1,40 | 4,2 | 16 | 4 | cast nephropathy | Yes | 1,3 | a |
| P010 | 53 | male | MM | IgA | Kappa | at diagnosis | 2,530 | unknown | 6 | 1,1 | 92 | 1 | Unknown | Yes | 1,1 | c |
| P011 | 45 | female | AL-Amyloidosis | – | Lambda | at diagnosis | 120 | unknown | 6,2 | 0,7 | 108 | 1 | Unknown | No | 0,8 | c |
| P012 | 66 | male | MM | IgG | Kappa | at diagnosis | 101 | unknown | 0,2 | 1,2 | 66 | 2 | Unknown | NA | 0,9 | c |
| P013 | 65 | male | MM | – | Kappa | at diagnosis | 11,000 | unknown | 3 | 1,5 | 50 | 3 | Unknown | No | 1,7 | c |
| P014 | 54 | male | MM | IgG | Lambda | after 3 courses of therapy | 2,2 | unknown | 0,1 | 1,1 | 76 | 2 | Unknown | No | 1,1 | c |
| P015 | 68 | male | MM | IgG | Kappa | at diagnosis | 4,380 | unknown | >6 | 1,5 | 46 | 3 | Unknown | Yes | 0,9 | c |
| P016 | 59 | male | MM | – | Kappa | at diagnosis | 1,150 | 17,5 | 3 | 1,6 | 45 | 3 | Unknown | No | 1,23 | c |
| P017 | 72 | female | MM | IgG | Kappa | at diagnosis | 1,380 | unknown | 4 | 0,67 | 90 | 1 | Unknown | No | 0,8 | c |
| P018 | 69 | male | MM | IgG | Kappa | at diagnosis | 369 | unknown | <0.05 | 1.13 | 66 | 2 | Unknown | NA | unknown | c |
| P019 | 56 | male | MM | IgG | Kappa | at diagnosis | 484 | unknown | <0.05 | 1.79 | 41 | 3 | Unknown | Yes | 1.4 | c |
| P020 | 64 | female | MM | – | Kappa | at diagnosis | 4,120 | unknown | 2.4 | 0.81 | 74 | 2 | Unknown | NA | 0.85 | c |

an ÄKTA pure chromatography system (GE Healthcare) using a Superdex 75 10/300 GL column.

### Analysis of the LC and HSA-ratio

The purity and LC content of the samples was evaluated based on a combination of SDS-PAGE gels and Western blots with antibodies against human kappa light chain, human lambda light chain or human serum albumin (HSA).

### Differential scanning calorimetry (DSC)

Thermal unfolding of the various light chains was studied using a MicroCal VP-DSC instrument (Malvern, UK) by performing temperature ramps on the LC solutions from 10 °C to 90 °C with a heating rate of 0.8 °C/min. The degree of refolding was estimated by the ratio of the areas under the unfolding peaks of the second to the first temperature scan.

### Determination of the dimer content

To determine the ratio of dimers to monomers of the various LC's, the proteins were run, before purification via SEC, on denaturing, non-reducing SDS-PAGE gels, which were stained using Coomassie blue.

### Proteolysis

LCs at an estimated concentration of 33 µM were incubated with bovine trypsin (molar ratio 1:100) at 37 °C in 10 mM phosphate buffer, pH 7.5 with 1 M urea. The preparation was done on ice and the aliquot for the first time point was collected after trypsin addition and immediately inhibited by adding trypsin inhibitor (at an excess of 2:1). Further aliquots were taken 1, 2, 18, 24 and 48 h after the addition of the trypsin and quenched by the addition of trypsin inhibitor. The samples were analyzed by SDS-PAGE.

### Amyloid fibril formation in vitro

Different solution conditions (pH, trypsin) were tested for their potential to induce amyloid fibril formation of the patient-derived, purified LCs. The amyloid formation was monitored using 20 µM of the fluorescent dye Thioflavin-T (ThT) in a multi-well plate reader.

## RESULTS & DISCUSSION

### Clinical data

The results presented are based on the urine samples of 20 patients with a monoclonal light chain (14 Kappa and 6 Lambda) related to different types of multiple myeloma and one patient with confirmed amyloidosis. Common denominator for inclusion into this in vitro study was the availability of a sufficient amount and purity of protein extractable from the urine. Further experiments were then performed with those samples that contained a dominant proportion of LCs. As a consequence, two samples (P008 and P009) were excluded from further analysis, as they mainly contained HSA. The data for the evaluation of LC content in the samples can be found in the Supplementary Materials.

We have chosen this clinically unbiased approach (i.e., no exclusion of patients based on clinical phenotype) in terms of day-to-day practicability rather than applying restrictive inclusion criteria. By doing so, we expected a better evaluation of the utility and relevance

of the biochemical and biophysical methods within the diagnostic work flow. At the time of inclusion into the study, the majority of patients were newly diagnosed without previous therapy. Two of the patients required dialysis. By investigating a potential relationship between the LC concentration in the blood and the degree of renal impairment at the time of the assessment we found no statistically relevant relationship (correlation coefficient of 0.16 with $p$-value of 0.54).

We were also interested in the further course of the disease and examined the kidney looking for an improvement of the creatinine concentration in serum during the course of the treatment. Without adequate information on P005 and P013 it was interesting to note, that of the 18 patients available for assessment even patients of the category 3 showed an improved kidney function after treatment. On the other hand, the renal insufficiencies of P002 and P003 are probably irreversible. The kidney function of most of the patients of category 1 did not change significantly, as their kidney function at the time of examination was not or only moderately impaired. Previous studies have shown that patients with LCDD and active MM displayed a greater improvement of kidney function in comparison to pure LCDD (*Muzaffar et al., 2017*).

## Thermal stability

It has been proposed that the thermodynamic stability of a light chain correlates with its tendency to form amyloid fibrils (*Blancas-Mejía et al., 2015*; *Hurle et al., 1994*; *Klimtchuk et al., 2010*; *Raffen et al., 1999*; *Del Pozo Yauner et al., 2008*), but this correlation has not been observed in all studies (*Bernier & Putnam, 1963*). Here we tested whether the stability against thermal unfolding correlated with the severity of the disease symptoms, independently of the fact if amyloid fibrils or other types of aggregates cause those symptoms. The thermal stability of the proteins was investigated using differential scanning calorimetry (DSC). The transition temperatures ($T_m$) of the different samples range from 50 °C to nearly 70 °C (Fig. 1B). In contrast to the other samples, which displayed one distinct peak for the unfolding of the LC, the thermogram of P004 showed two peaks; hence the presented $T_m$ 57.7 °C is a mean of the $T_m$ values of both peaks (53.2 °C and 62.1 °C).

The mean unfolding temperature of the kappa LCs of this study is 53.7 °C and the average unfolding temperature of the lambda LCs is 58.6 °C. Lambda LC have on average a greater stability against thermal unfolding, which is in agreement with published reports (*Bernier & Putnam, 1963*). No correlation was observed between the thermal stability and the severity of the patient's kidney function.

It has been reported that LCs derived from MM-patients, in contrast to those from AL-patients, were not able to refold after heat denaturation (*Andrich et al., 2017*). Therefore it has been speculated that this experimental parameter could be predictive for the in vivo behavior. In order to investigate the degree of reversibility of the thermal unfolding, the samples were heated for a second time, following a first ramping from 10 °C to 90 °C and subsequent cooling down within the DSC liquid cell. If the protein can quantitatively refold into the native conformation after thermal unfolding, the thermograms of the two successive scans should not differ significantly. If only a fraction of the protein can refold
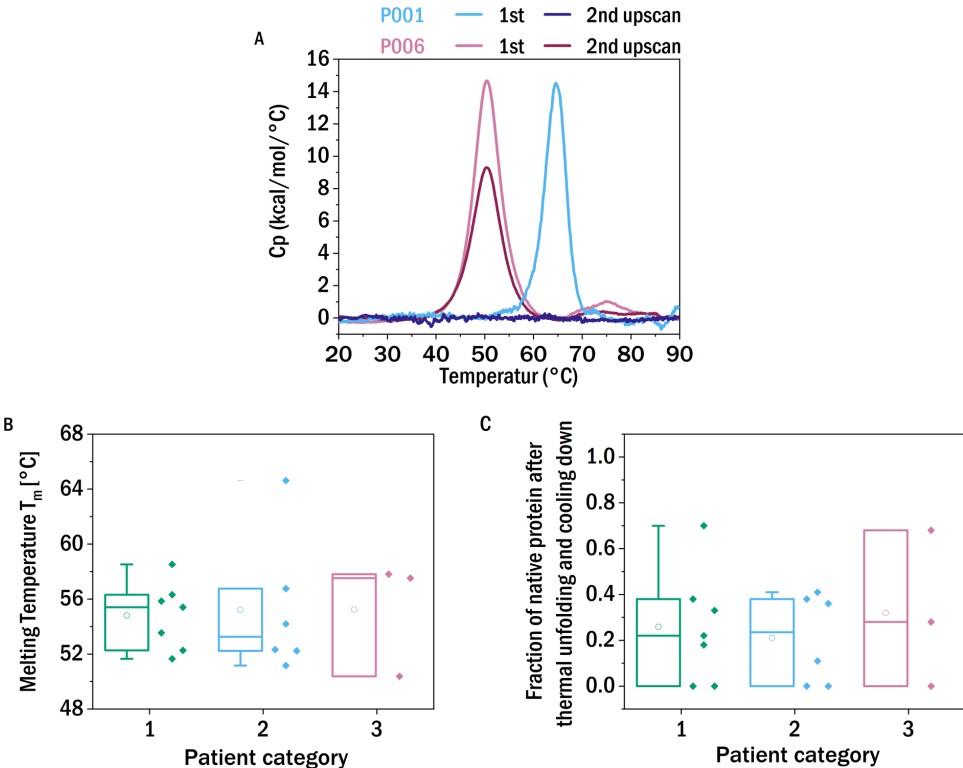

**Figure 1** **Thermal stability and reversibility of unfolding.** (A) Thermograms of the first and second DSC scans of P001 and P006. (B) Unfolding temperatures determined from differential scanning calorimetry (DSC) experiments of the different samples allocated in the three patient categories according to renal impairment (CDK stage): I (green), II (blue), III (red) (left). (C) The fraction of native protein that unfolds during the second scan. The boxes range between 25 and 75 percent, the median is visualized by the horizontal line and the mean by the small square. P008, P009 (low LC content), P014 and P018 (low sample availability) are excluded from this analysis.

into the initial structure, and the remaining fraction misfolds or aggregates, the integral of the DSC thermogram, i.e., the enthalpy of unfolding, correspondingly decreases or even vanishes. This decrease in the apparent enthalpy of unfolding is caused by the fact that only a correctly folded protein has a well-defined unfolding transition, whereas misfolded or aggregated protein comprises a multitude of states, often highly stable, that do not undergo a thermally induced transition at a well-defined temperature.

The observed degree of refolding upon reheating up to 90 °C was highly variable. Some LCs did not display a peak in the thermogram at the second temperature ramp, whereas other examined LCs show a high reversibility (up to 70%) of folding (Fig. 1C). Again, we found no apparent correlation between the ability of refolding after heating to 90 °C and the patients' renal function. Although of more fundamental interest, we also tested for a correlation between thermal stability and refolding ability of the LCs (Fig. S5) and found a weak inverse correlation, i.e., the more thermostable a given LC, the smaller its ability to refold. We discuss possible origins of this connection in the SI.
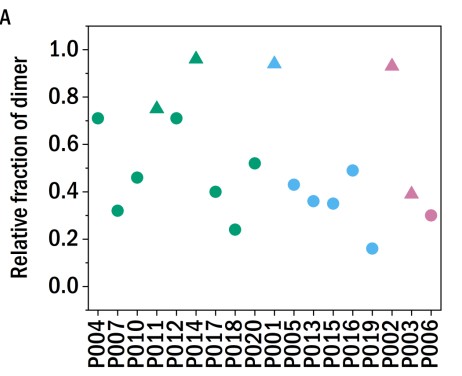 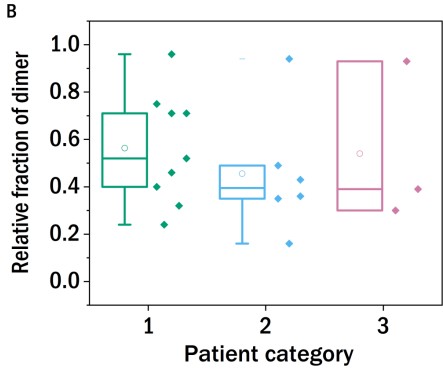

**Figure 2** **Monomer-dimer distribution of the LCs.** The relative fraction of dimer of the different samples measured in relation to the overall amount of native light chains (monomer and dimer), as determined by SDS-PAGE. The colors refer to the corresponding patient category defined above: I (green), II (blue), III (red), the shape indicates the isotype of the light chain: triangle: lambda, circles: kappa.

## Dimerization

The conformation of pathogenic LC proteins is typically dimeric (*Bernier & Putnam, 1963*; *Epp et al., 1975*), and it has been proposed that the tendency to dimerize can be used as a diagnostic parameter for the tendency to form amyloid fibrils, whereby light chains in serum samples derived from AL patients show abnormally high levels of monoclonal free light chain (FLC) dimers on a western blot (*Gatt et al., 2018*). On the other hand, the formation of dimers seems to prevent the LC to aggregate in vitro. The formation of amyloid fibrils was observed when the dimer was destabilized and dissociated into monomers (*Brumshtein et al., 2014*; *Wolwertz et al., 2016*; *Nawata et al., 2017*).

In order to test whether the degree of dimerization can be used as a predictor for general disease severity in the form of kidney impairment in LC disease, we performed similar experiments in our study. The relative fraction of dimers of the LCs after precipitating the proteins from the urine, redissolving them in buffer and purification by dialysis was determined with a non-reducing SDS-PAGE gel and quantified photometrically (Fig. 2). The fraction of dimers is high in some of the LC, and none of the examined LCs occurs exclusively in its monomeric form. A clear correlation between the fraction of dimerized LC and the renal function of the patients was not observed. In the case of P011, the sample from the only patient with confirmed amyloidosis, the dimer fraction of 0.75 is in accordance with the findings of the FLC-MDPA test that amyloid prone LCs occur mainly in their dimeric forms (*Gatt et al., 2018*).

## Trypsin digestion

The structural flexibility and dynamics of the LC have been proposed to be correlated with the ability to form amyloid fibrils in vivo. LC derived from AL-patients displayed rapid proteolysis with trypsin under mildly destabilizing conditions which suggests increased dynamics of the native fold of the amyloid-forming LC (*Oberti et al., 2017*). These structural dynamics may be required to form amyloid fibrils under physiological conditions. In the

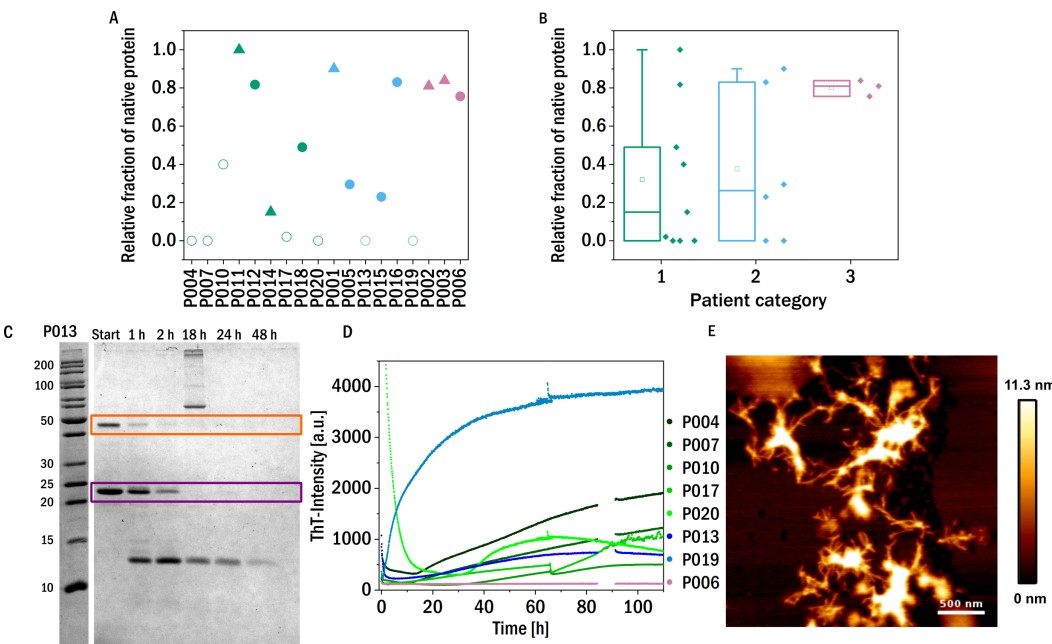

**Figure 3 Trypsin digestion.** (A) & (B) The fraction of native protein (monomer and dimer combined) after 48 h incubation with trypsin is displayed. The colors refer to the corresponding patient category defined above: I (green), II (blue), III (red). The shape of the symbols represents the isotype of the LC: triangles: lambda isotype, circles: kappa isotype. Empty symbols: amyloid fibril formation induced by proteolysis and inferred from an increased signal of ThT-fluorescence, filled symbols: no evidence for amyloid formation observed. (C) SDS-PAGE gel of the trypsin digestion of P013 as a representative example. The dimer (orange) and the monomer (purple) are marked with a square. (D) ThT fluorescence aggregation assay of the amyloid forming LCs. Data for P006 is added as an example of a LC that does not from amyloid fibrils. (E) AFM-image of the aggregated sample P007 after digestion with trypsin during the kinetic experiment in a multiwell plate.

present study, we performed such experiments in order to test whether structural flexibility of the light chains correlates with the severity of LC disease symptoms.

The structural flexibility and dynamics of the LC were probed by digesting the proteins using trypsin in the presence of 1 M urea. Fast proteolysis under these conditions may suggest increased dynamics in the native fold of the LC, which allows the protease to access the cleavage sites more easily. Of course it is important to remember that the sequence variability between different light chains will lead to different numbers and accessibilities of trypsin cleavage sites. The availability of sequence information of the investigated light chains would therefore be beneficial for the detailed analysis of their proteolytic susceptibility, and will be included in a future study. The digestion time course was analyzed using SDS-PAGE gels by determining how much native protein (monomer and dimer combined) was still present after 48 h incubation with trypsin (1:100 molar ratio) at 37 °C (Figs. 3A and 3B).

In addition to correlating trypsin digestion with kidney impairment, we also correlated it with the thermal stability of the LCs (Fig. S5). A weak inverse correlation is observed, i.e., the more thermostable a LC, the less it is susceptible to be degraded by trypsin. While this

connection is also of interest in a more fundamental protein science context, it does lend some support to the hypothesis underlying the digestion assay, see discussion in the SI.

These proteolysis experiments were in addition also conducted inside a multi-well plate reader in the presence of the fluorescent, amyloid specific dye ThT. Only seven of the tested samples showed an increase in fluorescence intensity indicating amyloid fibril formation (Fig. 3D and Fig. S2). Six of these seven amyloid positive LCs were almost completely digested after 48 h, suggesting that proteolytic cleavage of the LC facilitates the formation of amyloid fibrils. Proteolysis-induced amyloid fibril formation of light chains is a well-established phenomenon, and even the aggregation of non-amyloidogenic light chains after acidic proteolysis was observed (*Linke, Zucker-Franklin & Franklin, 1973*; *Glenner et al., 1971*). The presence of amyloid fibrils was confirmed by atomic force microscopy (AFM, Fig. 3E). Interestingly, the samples of patients of category three, i.e., the patients most severely affected by impairment of kidney function, were among the least digestible proteins tested. This observed behavior should be compared to the findings on the behavior of amyloid prone LCs previously reported (*Oberti et al., 2017*), whereby it was found that amyloidogenic LCs are most easily digested. Interestingly, the only established amyloidosis-related LC of our dataset (P011) was not digested during the investigated time period and was also not observed to form amyloid fibrils under the conditions of this experiment. However, the presence, in our data set, of only a single LC known to form amyloid fibrils in vivo did not allow us to test the previously reported correlation between proteolytic degradability and amyloid fibril formation.

On the other hand, while easy degradability might increase the risk for amyloid formation, our data set does suggest that LCs which are difficult to proteolytically degrade tend to be associated with impairment of the renal function. A putative mechanism by which this association can be explained is the more rapid accumulation, probably in the form of amorphous aggregates, of non-cleavable LC in the kidneys. At the same time, it has to be kept in mind that the proteases responsible for the degradation of LCs in vivo have cleavage patterns distinct from that of trypsin and therefore the digestibility by trypsin will only have limited predictive power for the digestibility of a given LC in vivo.

## pH-dependent amyloid formation

Given our findings that digestion by trypsin facilitated amyloid fibril formation of our patient-derived LCs, we further tested whether variations in solution conditions (pH, reducing agent) are also able to induce amyloid fibril formation in a subset of our samples. We found that at neutral pH in the presence of the reducing agent TCEP at 37 °C almost all the samples (except P002, P013, P014, P016, P020) show an increase in ThT fluorescence over time, suggestive of amyloid fibril formation, (see Fig. 4A for representative data and Fig. S4 for an overview). It is known that a reduction of mostly intermolecular disulfide bridges of light chains can induce amyloid formation (*Bernier & Putnam, 1963*). If the pH was lowered towards more acidic values (∼pH 4), fragmentation of the chains was observed, and in most cases amyloid fibril formation was absent at acidic pH in the presence of TCEP. Presumably, the degree of fragmentation of the LC through the combined effects
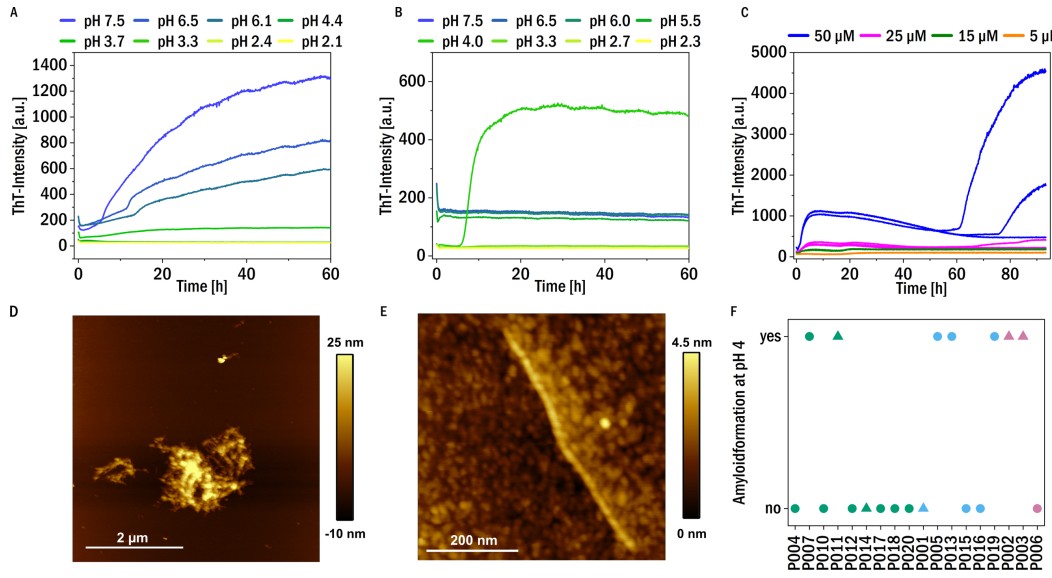

**Figure 4   Amyloid fibril formation by P005 under different solution conditions as a representative example.** (A) Aggregation assay at different pH values in the presence of 1 mM TCEP and (B) in the absence of TCEP. (C) Aggregation assay at pH 4 at different monomer concentrations measured through ThT-fluorescence. (D) and (E) AFM images of the sample P005 at 50 μM, at the end of the experiment shown in (C). (F) The amyloid fibril formation behavior at pH 4 of the LCs monitored by an increase in the fluorescence intensity of the dye ThT: yes: if the amyloid formation was observed; no: no evidence of amyloid fibril formation was observed. The colors refer to the corresponding patient category defined above: I (green), II (blue), III (red). The shape typify the isotype of the LC: triangles: lambda isotype, circles: kappa isotype.

of TCEP and low pH was too strong, such that the resulting short fragments were unable to form amyloid fibrils.

On the other hand, no amyloid fibril formation was observed at neutral pH in the absence of a reducing agent (see Fig. 4B). If, however, in the absence of reducing agent the pH was decreased, some of the LCs showed formation of amyloid fibrils at pH 4, in particular also P011, derived from the patient with confirmed amyloidosis (Fig. S3). Low pH destabilizes the proteins and mildly destabilizing conditions are known to accelerate aggregation (*Hu et al., 2008*). For example, investigations of the amyloidogenic variable domain SMA showed relatively native-like intermediates, but with significant changes of the tertiary structure, at pH 4 (*Khurana et al., 2001*). A summary of the aggregation behavior at pH 4, together with the corresponding patient categories of the samples, is given in Fig. 4F. Seven LCs formed amyloid fibrils at pH 4. Figure 4C displays a representative example of the observed aggregation kinetics of sample P005 at pH 4. The resulting amyloid fibrils were also in all cases examined using AFM imaging (see Figs. 4D and 4E for a representative example, and Fig. S3). The fibrils seem to form clusters/higher order assemblies at this pH, but also individual fibrils were detected. Some of the LCs displayed amyloid fibril formation only at pH 4 and did not form amyloid fibrils if the pH was decreased to pH 3 and below. Very low pH may lead to too substantial fragmentation of the LC (*Vlasak & Ionescu, 2011*), similar to the combined effect of reducing agent and less acidic pH discussed above. There is no

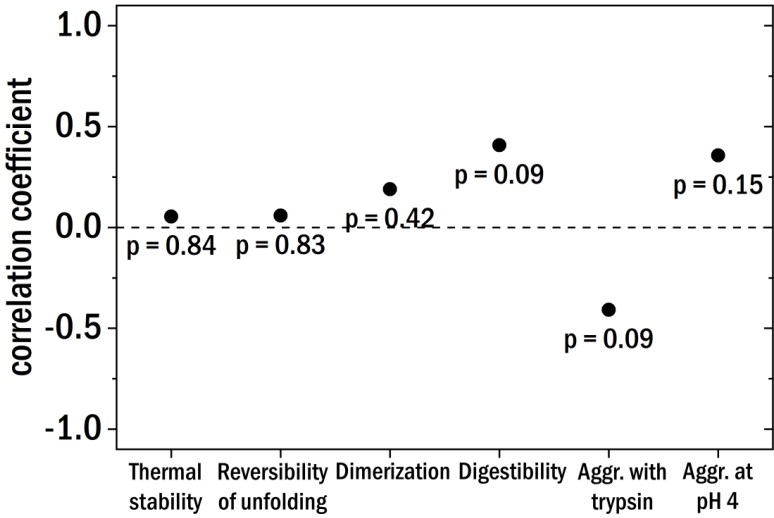

**Figure 5** **The Pearson correlation coefficient between the clinical patient categories and the different investigated biochemical and biophysical characteristics.** The points are labeled with the corresponding *p*-value.

obvious correlation observable between the impairment of kidney function at the time of diagnosis, i.e., the patient category, and the aggregation behavior under the investigated solution conditions.

## CONCLUSIONS

This study provides a comprehensive evaluation of the *in vitro* biophysical and biochemical behavior of patient-derived monoclonal light chains. The majority of patients had been diagnosed *de novo* with multiple myeloma, while a histopathological examination was performed in three patients leading to the diagnosis of cast nephropathy in two patients and AL-Amyloidosis in one. We used an unbiased group of patients as they came to our hospital during our recruitment phase. Therefore, different from recent studies reporting on patients with light chain disease we cannot make an *a priori* separation into two distinct groups on the basis of the amyloid formation behavior of the respective LC proteins. However, an *a posteriori* bias was introduced into our patient group by excluding two samples from further analysis, because they did not contain a sufficient concentration of light chains, but were dominated by albumin.

Starting with the thermal stability of the LCs and their ability to refold after thermal unfolding we could not find a clear correlation with the degree of kidney damage (see Fig. 5 for correlation coefficients). Also, the relative amount of dimer and the aggregation behavior at pH 4 were not correlated to the different patient categories. The digestibility by trypsin of the LCs yielded no clear correlation with kidney damage for our samples. According to published results (*Oberti et al., 2017*), LCs derived from AL-amyloidosis patients could be relatively easily digested whereas LCs from patients with multiple myeloma seemed to be more resistant against digestion by trypsin. Trypsin digestion is here used as a proxy for

structural dynamics, with the caveat that different LCs are likely to differ in the number and availability of trypsin cleavage sites.

The most noteworthy result of our study is that in our dataset, the LCs of the three patients with the greatest degree of renal impairment were only digested by trypsin to a very small extent. Therefore, while easily digestible LC may have a tendency to form toxic amyloid fibrils in vivo, our results suggest that indigestible LCs may nevertheless be able to induce severe kidney damage, due to their overall higher structural stability their resulting ability to accumulate in the kidneys. This conclusion will, however, need to remain somewhat speculative until sequence information of the LCs of this study becomes available (ongoing work), which will allow to investigate how physiologically more relevant proteases can digest these LCs.

The particular feature of LC diseases, namely that every patient displays a LC of a unique amino acid sequence, makes it necessary to substantially enlarge the currently available data set that links biophysical, biochemical and sequence information with clinical disease symptoms and the present study aims to contribute to this task. In order to obtain a better mechanistic understanding of the in vivo behavior of the LCs, a more detailed biophysical investigation, as well as sequence determinations (see above) of the extracted light chains, are required. These are the subject of ongoing studies of our groups.

### Funding
This work was supported by the Manchot Foundation and the Novo Nordisk Foundation (NNFSA170028392). The funders had no role in study design, data collection and analysis, decision to publish, or preparation of the manuscript.

### Grant Disclosures
The following grant information was disclosed by the authors:
Manchot Foundation and the Novo Nordisk Foundation: NNFSA170028392.

### Competing Interests
The authors declare there are no competing interests.

### Author Contributions
- Rebecca Sternke-Hoffmann conceived and designed the experiments, performed the experiments, analyzed the data, prepared figures and/or tables, authored or reviewed drafts of the paper, and approved the final draft.
- Amelie Boquoi performed the experiments, prepared figures and/or tables, authored or reviewed drafts of the paper, and approved the final draft.
- David Lopez Y. Niedenhoff performed the experiments, authored or reviewed drafts of the paper, and approved the final draft.
- Florian Platten, Roland Fenk, Rainer Haas and Alexander K. Buell conceived and designed the experiments, analyzed the data, authored or reviewed drafts of the paper, and approved the final draft.

## Human Ethics

The following information was supplied relating to ethical approvals (i.e., approving body and any reference numbers):

The ethics committee of the university hospital Düsseldorf granted ethical approval to carry out the study with urine samples from patients (study number 5926R and registration ID is 20170664320).

All patients of whom samples were used have signed an informed consent.

## Data Availability

The raw data is available at the Open Science Framework: Buell, Alexander K, and Rebecca Sternke-Hoffmann. 2019. ''Patient/derived Ig Light Chain Characterisation.'' OSF. October 23. osf.io/4r8mx.

## Supplemental Information

Supplemental information for this article can be found online at http://dx.doi.org/10.7717/peerj.8771#supplemental-information.

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
