# Peer review of "Biochemical and biophysical characterisation of immunoglobulin free light chains derived from an initially unbiased population of patients with light chain disease"

_PeerJ, doi:10.7717/peerj.8771_

## Round 0.1 · original submission · Minor Revisions

We apologize for the delay in the decision. The manuscript has been now evaluated by two independent reviewers who agree on minor revisions. Please refer to their comments attached, we would be glad to consider a version in which the points have been addressed one by one.

Reviewer 1 ·

Basic reporting

The manuscript by Sternke-Hoffmann et al. characterizes the purity, thermal unfolding, tryptic digestion and aggregation kinetics of a set of light chain proteins from patients suffering from multiple myeloma (MM) and / or light chain (AL) amyloidosis. The study finds no correlation between protein stability, aggregation kinetics and the clinical phenotype of the disease, specifically the renal impairment. There was, however, an increased resistance to tryptic digestion in LC from patients with high renal impairment. Unfortunately, possible mechanistic explanations of this observation remain largely unexplored.

As the manuscript essentially reports a data set, the reported data should be as complete as possible. I would strongly suggest putting the original data from AFM, aggregation assays and Western blotting from all 20 samples into the supplementary data set. This would enable the scientific community to mine the reported data set for future studies. The authors have already deposited SDS-PAGE and DSC data for all samples, which is commendable, but AFM data and ThT kinetics seem to be missing for most samples. I would also suggest adding a meta file to the supplements that lists explicitly, which data files contain which experiments and samples.

Experimental design

The main benefit of this paper to the scientific community is the thorough characterization of a LC protein sample set. The data presented are solid and I see no flaws in experimental design. Therefore type of manuscript appears to be within the scope of PeerJ, even though it leaves the reader asking which, if any, hypothesis was being tested.

The main methodical criticism is that information about the criteria for patient selection is lacking. This is important, since all the LC samples in the study came from patients with a high ratio of LC to albumin in the urine. While this is common in MM, only a small subset of AL patients fulfills this criterion, as many patients present with massive proteinuria, leading to the excretion of large amounts of albumin and comparatively little LC, which would have prevented the purification strategy employed in this study. The samples in this study may therefore not be representative of MM and AL cases in general. This apparent and unstated bias needs to be resolved, especially since the title emphasizes the ‘unbiased’ population of patients. The authors need to report whether the sample set simply consisted of a random set of MM patients, one of whom happened to be diagnosed with AL, or whether biochemical criteria about LC / albumin ratios were applied during patient selection and, if so, how their patient selection relates to the overall population of light chain disease patients.

Validity of the findings

The findings as reported are valid and the experimental design seems robust. Criticism and suggested improvements are stated in the previous two sections. It would be helpful to know if the data set in this manuscript will form the basis of further mechanistic or therapeutic studies or whether the authors are reporting a negative result based on a specific hypothesis.

Additional comments

Figure formatting should be improved before publication. Axis labels are uneven between panels. Many of them are much too small to be legible (Fig. 3 and 4). The AFM panel in Fig 3 lacks height scale.

Reviewer 2 ·

Basic reporting

The paper “Biochemical and biophysical characterisation of 1 immunoglobulin free light chains derived from an unbiased population of patients with light chain disease” presents some findings on the biochemical properties of antibody light chains from patients with light chain disease, either diagnosed with MM or AL
BASIC REPORTING
The paper is well written and very easy to understand. It gives a fair description of the work in the field and the references are sufficient. My only suggestion is to remove the acronym CNS since it is not defined elsewhere and only used once.
All the findings are also displayed through a few, quite clear pictures. In some cases the pictures could be improved: the dark green/light green boxes can be done in different colors. Also, the different properties are always displayed individually, and never combined in a single plot. For example, it should be expected, according to what stated in the paper, that the thermal stability and the tryptic digestion values might have some correlation. The paper never integrates or discusses the different features altogether – see later for more details
Also, no raw data is supplied

Experimental design

EXPERIMENTAL DESIGN
See the "validity of the findings" sections about the experimental setup for digestibility.
All experiments are performed and described in a rigorous and replicable fashion

Validity of the findings

VALIDITY OF THE FINDINGS
Most of the results in this work are solid, even if negative or incremental. I would have liked to see if some of the features have a correlation, such as thermal stability and digestibility. Since the digestibility is primarily a way to measure the extent of dynamics of the LCs, it might be expected that these features correlate to a certain extent.

I am also very unsure that digestibility as presented is a very solid way to measure the extent of protein dynamics across different LCs. Trypsin has a very precise cleavage pattern, and the number of tryptic sites can vary hugely between different LCs. It would be extremely beneficial to also sequence the LCs from the patients’ B cells. This would have not only given an answer to e.g. the different number of tryptic sites, but also a valuable information on the precise differences among the patients’ LCs.
I would suggest that the authors comment on these points.

On the other hand, the digestibility is in some parts argued to be a possible starting point for fibrillation. This is true, but in my understanding the proteases that are involved in this process in vivo are completely different from trypsin, so these findings might not reflect to the actual digestibility of the protein in vivo. I would suggest that, if the authors want to study digestibility per se and not as a measure of protein dynamics, they replicate the experiment with the proteases that are actually involved in LC proteolysis.

Additional comments

The paper is clear and well written, and I believe that some of the findings are of interest in the field. I believe that some of the conclusions are not completely supported by the experiments, and they are mostly of speculative nature. I would therefore suggest to discuss these conclusions more thoroughly

---

## Round 0.2 · accepted · Accept

I am really glad to endorse this manuscript for publication. All the questions from the reviewers have been nicely and properly addressed.

Reviewer 1 ·

Basic reporting

In their revised manuscript, the authors have comprehensively addressed my comments and submitted a manuscript that is ready for publication.

Experimental design

no further comment.

Validity of the findings

The authors have expanded the data deposited at OSF to provide comprehensive access to their primary data.

Reviewer 2 ·

Basic reporting

The manuscript in the current form is suitable for publication

Experimental design

no comment

Validity of the findings

no comment